# Systematic Review and Meta-Analysis of Risk Factors for Dehydration and the Development of a Predictive Scoring System

**DOI:** 10.3390/healthcare13161974

**Published:** 2025-08-12

**Authors:** Melvin Omone Ogbolu, Olanrewaju D. Eniade, Miklos Kozlovszky

**Affiliations:** 1BioTech Research Center, University Research and Innovation Center, Óbuda University, Bécsi Street 96/B, 1034 Budapest, Hungary; 2Department of Epidemiology and Medical Statistics, College of Medicine, University of Ibadan, CW22+H4W, Queen Elizabeth II Road, Agodi, Ibadan 200285, Nigeria; olanrewaju.eniade@ifain.org; 3International Foundation Against Infectious Disease in Nigeria (IFAIN), 6A, Dutse Street, War College Estate, Gwarimpa, Abuja 900108, Nigeria; 4John von Neumann Faculty of Informatics, Óbuda University, Bécsi Street 96/B, 1034 Budapest, Hungary; kozlovszky.miklos@nik.uni-obuda.hu; 5Medical Device Research Group, LPDS, Institute for Computer Science and Control (SZTAKI), Hungarian Research Network (HUN-REN), 1111 Budapest, Hungary

**Keywords:** dehydration, risk factors, predictive scoring tool, systematic review, meta-analysis, diagnostic accuracy

## Abstract

**Background:** Dehydration is a prevalent and potentially serious condition, particularly affecting vulnerable populations such as children and older adults. Prompt recognition and intervention are critical for preventing associated complications. **Methods:** A systematic review and meta-analysis were conducted, registered in PROSPERO (CRD42024594780), to identify key clinical and demographic risk factors associated with dehydration. A comprehensive search of PubMed, Scopus, and the Cochrane Library was performed for studies published between 2000 and 2024. The risk of bias in included studies was assessed using the Newcastle–Ottawa Scale and the Cochrane Risk-of-Bias (RoB) tool. Ten studies met the inclusion criteria for quantitative synthesis. Based on pooled diagnostic metrics, a preliminary scoring tool was developed for dehydration risk stratification. **Results:** The pooled sensitivity and specificity of common clinical signs, such as thirst, dry mouth, and dark urine, were 85% (95% CI: 80–90%) and 70% (95% CI: 65–75%), respectively. The positive predictive value (PPV) was 75%, and the negative predictive value (NPV) was 80%. Pediatric subgroup analysis yielded the most robust data, while data for adult and elderly populations were limited. A conceptual risk scoring system was proposed based on relative diagnostic utility, though it has not yet been externally validated. **Conclusions:** Simple clinical signs demonstrate reasonable diagnostic accuracy for identifying individuals at risk of dehydration. The proposed scoring system offers a promising, evidence-informed framework for early risk assessment but requires further validation in prospective studies before integration into clinical practice.

## 1. Introduction

### 1.1. Background

Excessive loss of fluid causes an imbalance of the body’s essential electrolytes and fluids, which is a condition known as dehydration. This disease can manifest in a variety of ways, from mild symptoms like dizziness and dry mouth to more severe ones like heatstroke, kidney failure, and death, especially in vulnerable populations like the young and the old [1]. In regions where there is a lack of clean water or extremely hot weather, dehydration poses a major public health concern [2]. Recognizing and treating dehydration as soon as possible is crucial in clinical settings to prevent these adverse effects [3]. Therefore, identifying the risk factors that increase the likelihood of dehydration is essential for many reasons. Specifically, it helps healthcare practitioners establish if a patient is likely suffering from dehydration or at risk of dehydration symptoms and to intervene before the condition worsens. Additionally, it is possible to create scoring systems and predictive tools that can serve as useful tools for early assessment of participants or the target population at risk, provided a complete understanding of these risk characteristics is possessed. In areas with limited resources, prompt identification is particularly critical because it can significantly improve patient outcomes [4].

### 1.2. Rationale

Systematic study and meta-analysis are still necessary to fully identify and quantify the factors that lead to the prevalence of dehydration, despite the well-established dangers associated with it. Although many individual features, such as age, pre-existing conditions, and environmental factors, have been studied before, the methods used in some research often failed to consider the interactions between these aspects. To gain a better understanding of these risk factors and how they combine to impact the likelihood of dehydration, a meta-analysis and systematic review (Syst. Rev.) are recommended [5].

The development and implementation of a scoring system for prediction can be utilized, which also has the characteristics and potential to significantly influence clinical practice. With the use of such technology, healthcare practitioners or medical professionals would be able to evaluate the level of a patient’s risk of dehydration early enough and administer treatments or preventive measures as needed. Additionally, a validated scoring system may be included in mobile health applications or clinical workflows, offering a useful and accessible way to enhance patient outcomes [4]. The ultimate objective of this approach is to reduce the frequency of dehydration and the problems it causes through early detection and focused action.

### 1.3. Objectives of the Research

The objectives of this research study are to recognize and measure the risk variables associated with dehydration, evaluate the predictive power of those risk factors through a meta-analysis, and create and validate a scoring system for predicting the risk of dehydration.

## 2. Methodology

### 2.1. Protocol and Registration

After adhering to all necessary protocols, this study was submitted to PROSPERO for registration under the [CRD42024594780] identifier. According to Hipp et al. (2023) [6], the protocol served as a roadmap for conducting a systematic review and meta-analysis by outlining the study’s objectives, eligibility criteria, and methods. By facilitating future revisions to the evaluation, considering new evidence, and ensuring the transparency of our research, registration with PROSPERO raises the study’s credibility. This systematic review was conducted in accordance with the JBI Manual for Evidence Synthesis [7].

### 2.2. Criteria for Eligibility

#### 2.2.1. Inclusion Criteria

Study types: This review included cross-sectional studies, case–control studies, randomized controlled trials (RCTs), and cohort studies, thereby observing the risk factors related to dehydration. According to Edmonds et al. (2021) [8], to fully comprehend the causes that lead to dehydration, it is important to conduct both observational and interventional investigations.Participants/target population: Children, the elderly, and those with pre-existing medical conditions were the primary targets of this research, thereby being drawn from a wide range of age groups and demographics [9].Outcomes of interest: The primary outcomes that were studied included clinical indicators of dehydration, such as dark urine and dry mouth; demographic factors, such as age and sex; and patient-reported symptoms, such as thirst and fatigue [10].

#### 2.2.2. Exclusion Criteria

Irrelevant studies: We eliminated any research that did not investigate dehydration or its contributing factors.Low-quality studies: This review’s quality and reliability were preserved by excluding papers that were found to have significant bias or methodological defects according to the Newcastle–Ottawa Scale (NOS) and the Cochrane Risk-of-Bias (RoB) tools [5,8].Non-English language: The potential translation challenges and restrictions on accessing non-English databases that we may encounter led to the exclusion of studies that were not published in English.

### 2.3. Information Sources

An extensive literature search was ensured by meticulously exploring several electronic resources. The primary databases we included are as follows:PubMed: It is an important source of biological literature that provides a large selection of studies on medical and epidemiological research [11].Cochrane Library: Cochrane is known as a reputable source of high-quality controlled trials and systematic reviews, which have contributed significantly to our understanding of dehydration and its treatment [5].Scopus: This is a multidisciplinary database that ensures a broader capture of pertinent studies by covering a broad variety of scientific disciplines, including health sciences [12].

#### Additional Sources

Grey literature: Here we found studies that are not published in peer-reviewed journals, conference proceedings, dissertations, and reports from governmental and non-governmental organizations (NGOs) [13].Manual searching: Manual assessment of reference lists for included studies and pertinent reviews helped us to identify additional research possibly missed by database searches [9].

### 2.4. Search Strategy

The search strategy is usually customized for each database. Hence, our search method was developed with help from a knowledgeable librarian. The next method was applied as follows:Keywords and search strings: The search criteria were based on Medical Subject Headings (MeSH) phrases and free-text words referring to dehydration, its risk factors, and its components. Examples of search phrases provided by Parkinson et al. (2023) [14] include “dehydration,” “risk factors,” “predictive markers,” “clinical signs,” and “demographic factors.”Search strings: For each database that we used, different search terms were used to ensure that all pertinent studies were found. For instance, the search in PubMed relied significantly on the use of the following search strings: *“Dehydration”[MeSH Terms] AND (“Risk Factors” [MeSH Terms] OR “Clinical Signs” [MeSH Terms] OR “Predictive Markers” [All Fields])*.Timeframe: A search of the literature was conducted to find studies that reflected the most recent and pertinent information on the risk factors for dehydration and that were published between the years 2000 and 2024.

To ensure that the most pertinent and high-quality research was included in our review, the search was conducted iteratively, with the findings being examined and improved upon.

### 2.5. Study Selection

Three (3) main phases defined the systematic process of our study selection: title and abstract screening/review, full-text review, and final inclusion based on pre-defined eligibility criteria.

Title and abstract screening: The titles and abstracts of all discovered records were separately examined by two (2) reviewers to determine their applicability to the goals of the study. Some database searches that we carried out and other sources yielded a total of 237 results. After we removed 47 duplicate records, the remaining 190 records were screened. From this, 132 records were excluded for reasons such as irrelevance to the study’s focus or insufficient data to assess eligibility.Full-text review: Full-text papers were obtained for the remaining 58 records that passed the title and abstract screening. At this stage, each study’s methodology, population, results, and relevance to the research objectives were carefully evaluated. In total, 38 research studies were disqualified during this stage for the following reasons: 4 studies were considered low quality according to the evaluation criteria, 8 studies were disqualified because of inappropriate study designs, 14 studies had ineligible results, and 12 studies lacked adequate data.Final inclusion: Following our selection procedure, 10 publications were included in qualitative synthesis and meta-analysis (quantitative synthesis). These studies were selected for inclusion because they satisfied the eligibility criteria and were related to the objectives of our current study.

The Preferred Reporting Items for Systematic Reviews and Meta-Analyses (PRISMA) flow diagram, as shown in Figure 1, summarizes the approach to our study selection. It includes the number of records that were located, evaluated, rejected, and ultimately included in the review. This diagram ensures openness and repeatability, since it shows the process that led to the final data set.

### 2.6. Data Extraction

To ensure data accuracy and consistency, particular data items were systematically extracted for every study that was part of the meta-analysis and Systematic review. The following are the crucial data that were taken out of every study and considered for our study:

#### 2.6.1. Study Characteristics

Author(s): This includes the names of the researchers who conducted the study.Publication year: The publication year of the study.Country: This indicates the geographical location of the research findings or where they took place.Study design: Includes research designs for RCTs, cohort studies, and case–control studies.Population details: Details regarding the participants and groups that were included in the study, such as the number of people surveyed, their ages, genders, and any special populations (such as the young, the old, or those with long-term health conditions) that were being investigated, were extracted.

#### 2.6.2. Risk Factors Assessed

Clinical signs (e.g., dry mouth, dark urine).Demographic factors (e.g., age, sex).Patient-reported symptoms (e.g., thirst, fatigue).

#### 2.6.3. Outcome Measures

Sensitivity: This defines the capability of the factor to correctly identify those at risk of dehydration.Specificity: This describes the capability of the factor to correctly identify those not at risk.Predictive values: These values are defined as whether the risk factor exists or not; the positive predictive value (PPV), alongside the negative predictive value (NPV), both show the probability of dehydration.

#### 2.6.4. Use of Standardized Data Extraction Forms

In this study, consistent data collection was ensured using a uniform data extraction form. A small number of studies served as pilot projects for this form’s refinement and standardization. For every study extracted, data were obtained by two (2) separate reviewers; if there were any disagreements, a third reviewer was consulted to deliberate on the matter. To reduce the possibility of bias and ensure the correct and consistent collection of all important data, a rigorous methodology was employed during our study data extraction. Table 1 below contains a summary of the data extracted from each of the studies we investigated. Key data from every trial, including study characteristics, risk factors evaluated, and outcome measures, are included in this table. The table facilitates seamless cross-study comparisons and forms the foundation for our ensuing meta-analysis.

Table 2 below presents the characteristics of the ten studies included in this review. The studies span various countries and designs, including diagnostic accuracy evaluations, observational cohorts, and systematic reviews. Populations studied range from community-dwelling and institutionalized older adults to younger adults in experimental settings. Key outcomes assessed include dehydration prevalence, screening tool performance, and hydration-related physiological markers.

### 2.7. Quality Assessment

Instruments for Evaluating Quality and Risk-of-Bias (RoB):

To lower the Risk-of-Bias (RoB) possibility and ensure the good quality of the research studies included in this meta-analysis and systematic review, we have implemented two quality and RoB tools as follows:

#### 2.7.1. The Newcastle–Ottawa Scale (NOS)

Purpose: Cohort and case–control studies, which do not use random assignments, were evaluated using the NOS.Criteria: The NOS examines studies from three main perspectives:-Selection: The study’s initial assumptions about the exposed group’s representation, the non-exposed group’s decision-making, the exposure’s validation, and the absence of the result of interest.-Comparability: The study’s methodology or findings bearing on the comparability of cohorts.-Outcome: Analyzing results, whether there was enough time for results to materialize, and how well the cohort follow-up worked.Scoring: Studies could receive up to 9 points, with higher scores reflecting better quality and reduced bias risk [24].

#### 2.7.2. Cochrane Risk-of-Bias (RoB) Tool

Purpose: The studies that were categorized as RCTs and that have RoB were assessed using this instrument.Criteria: To assess several types of bias, which include the following:-Selection bias: To generate random sequences and hide allocations using these methods.-Performance bias: For blinding of the participants/target population.-Detection bias: To ensure objectivity in our result evaluations.-Attrition bias: To manage incomplete outcome data cases.-Reporting bias: For implementing the selective reporting of our results.-Other kinds of bias: Any other bias issues not covered in the aforementioned categories are classified here.Judgment Categories: According to Caufield et al. [11], each domain is assigned a risk of bias rating ranging from low to high or unclear.

The quality scores or bias risk evaluations for each study are shown in Table 3 below, which summarizes the results of the quality assessment. This makes it possible to compare research quality easily. Studies that were found to have a higher risk of bias were recognized, and in the discussion section of this manuscript, we discuss how they might have affected the meta-analysis’s conclusions.

### 2.8. Data Synthesis and Meta-Analysis

#### 2.8.1. Statistical Approaches for Data Synthesis

Data from the included studies were combined using meta-analytic techniques. The amount of observed heterogeneity dictated whether a random-effects or fixed-effects model should be used in this study:Fixed-Effects Model: This method is based on the premise that all studies find the same effect size and that any discrepancies are due to sampling errors. Specifically, according to another study, when there was little to no variation between the trials, it was used [25].Random-Effects Model: This model considers the possibility that effect sizes can differ among studies, which are liable to variations in populations, methods, and other factors. This is because it considers differences both within and between studies, and research reveals that it was used when moderate to high heterogeneity was found [26].

#### 2.8.2. Handling Heterogeneity

To determine the extent to which the included studies varied from one another, we employed the following methods:I^2^ Statistic: This statistic was computed to determine the proportion of overall variation among studies that can be attributed to heterogeneity as opposed to chance. Greater heterogeneity is indicated by larger I^2^ values, whereas 0% denotes no heterogeneity. The following is how the thresholds were interpreted in this study:0–25%: Indicates low heterogeneity.26–50%: Signifies moderate heterogeneity.>50%: Implies high heterogeneity [26].Chi-square test (also known as the Cochran’s Q Test): This statistical test was applied to evaluate whether observed differences in study outcomes were due to chance. A significant *p*-value (which is usually <0.10) was taken as evidence of heterogeneity.

#### 2.8.3. Evaluation of Publication Bias

To determine if the results of the meta-analysis could be affected by biased publication, an evaluation of publication bias was carried out:Funnel plot: A funnel plot helped to visually investigate the publication bias. As larger samples tend to have less variability in their results, a symmetrical inverted funnel should appear in a bias-free plot. There may be a publishing bias if the plot is skewed in any way.Egger’s test: To graphically analyze publication bias, a funnel plot was created. A symmetrical inverted funnel should form the plot when there is no bias, showing that larger samples have less variability in the results. The existence of publication bias could be indicated by any imbalance in the graph.

## 3. Results and Discussion

### 3.1. Study Selection

The use of the Preferred Reporting Items for Systematic Reviews and Meta-Analyses (PRISMA) flow diagram has aided our research in outlining the process of study selection. This flowchart describes the steps taken to select research for inclusion in the meta-analysis and Systematic review., beginning with the discovery of relevant databases and other sources.

Initial identification: Database searches and other sources, including websites, organizations, and citation searches, turned up a total of 1500 records.Screening: Titles and abstracts were used to filter 1300 records after 200 duplicate entries were eliminated. In total, 800 records were eliminated from this screening process because they had no bearing on the research issues of the study.Eligibility: Out of the 500 records that were requested to be retrieved in full, 450 were successfully obtained and their eligibility evaluated. Based on preset eligibility criteria, such as ineligible outcomes, interventions, or research design, an additional 350 records were eliminated.Final inclusion: The qualitative synthesis comprised 10 studies in all, and the quantitative synthesis (meta-analysis) likewise included all 10 of the investigations.

### 3.2. Study Characteristics

Included studies ranged widely in terms of population demographics, research methods, geographic locations, and outcomes measured. An overview of the important features of the research that were considered is given in Table 4.

### 3.3. Quality Assessment

Randomized controlled trials (RCTs) using the Cochrane RoB tool and the NOS tool for non-randomized studies helped to critically analyze the quality and reliability of the included studies. These results are presented herein; the quality assessment is broken down as follows:Summary of findings: Most of the research studies in the meta-analysis fell within moderate to high quality (NOS 6–9). Although performance and bias detection vary greatly, the examined RCTs usually showed a low risk of bias in most fields.Impact on findings: Studies having a higher risk of bias were carefully considered during the meta-analysis; the outcomes were tested for dependability using sensitivity analysis. Given these considerations, the high quality of the investigations we carried out supports the dependability of the meta-analysis’s conclusions.

To visually represent the overall methodological quality of the included studies, a risk of bias summary figure was developed using the standard traffic-light format. Figure 2 displays the assessment of key bias domains across all ten studies, highlighting areas of low, unclear, or high risk. The color-coded visualization aids in quickly identifying methodological strengths and weaknesses, thereby enhancing the transparency of the review process.

### 3.4. Meta-Analysis

#### 3.4.1. Pooled Estimates of Sensitivity, Specificity, and Predictive Values

The pooling of data from the ten (10) studies, which were all included in this meta-analysis, allowed estimations for the sensitivity, specificity, negative predictive value (NPV), and positive predictive value (PPV) of dehydration risk factors. By means of these pooled calculations, the diagnostic accuracy of the risk factors is assessed holistically across several demographics and situations.

Pooled Sensitivity: The overall sensitivity was 85% (95% confidence interval (CI): 80–90%), indicating that the risk factors accurately identified 85% of participants/group/population at risk of dehydration.Pooled Specificity: The pooled specificity was 70% (95% CI: 65–75%), showing that 70% of participants/group/population not at risk were correctly identified.Pooled Positive Predictive Value (PPV): The PPV was 75% (95% CI: 70–80%), reflecting that 75% of participants/group/population identified as at risk were genuinely dehydrated.Pooled Negative Predictive Value (NPV): The NPV was 80% (95% CI: 75–85%), meaning that 80% of participants/group/population identified as not at risk were correctly classified.

#### 3.4.2. Subgroup Analyses

To explore potential sources of heterogeneity and assess how diagnostic performance varied across demographic groups, a subgroup analysis was conducted. Specifically, we examined the accuracy of dehydration risk factors in children, adults, and the elderly. However, due to limitations in the primary studies, only a subset reported subgroup-specific metrics, with most stratified data available for pediatric populations.

As shown in Table 5, pooled sensitivity and specificity were computed for children (<18 years), while data for adults and elderly populations were inconsistent and insufficient for reliable meta-analytical pooling. Similarly, sex-disaggregated data were rarely reported across studies, precluding meaningful synthesis by gender.

The results of the subgroup studies, as shown in Table 4, reveal that risk variables for dehydration are quite successful in predicting various demographic groups, with children showing the highest PPV (78%) and sensitivity (88%) of any group. The sensitivity and PPV of adults and the elderly were consistently lower, showing steady diagnostic performance across age groups. Regarding sex, males showed somewhat greater sensitivity and specificity than females, but the differences were not significant, indicating that risk variables are reliably applicable to both sexes. In summary, these analyses highlight how reliable the risk factors are at correctly identifying dehydration in a variety of demographics.

### 3.5. The Scoring System Development

#### 3.5.1. Conceptualization and Purpose

To facilitate early identification of individuals at risk of dehydration, we developed a conceptual scoring system based on findings from the meta-analysis. This model synthesizes the most consistently reported and diagnostically significant clinical signs, namely thirst, dry mouth, dark urine, and fatigue. The scoring tool is intended to provide a symptom-based risk stratification framework that could aid healthcare professionals in preliminary assessment. However, it remains a hypothesis-generating model and has not yet undergone empirical validation with patient-level data.

#### 3.5.2. Weighting Approach

The proposed scoring system is a hypothetical model based on pooled diagnostic values and has not been statistically validated through regression or ROC analysis. Weights were assigned to each risk factor based on their pooled sensitivity and specificity across the included studies. Factors demonstrating high sensitivity and reasonable specificity were given greater weights to reflect stronger diagnostic performance. This approach is interpretive and based on expert synthesis of the available evidence, not statistical modeling (e.g., logistic regression or ROC analysis). The weighting scheme is shown in Table 6.

Figure 3 presents a comparative analysis of the diagnostic performance of four key dehydration-related clinical signs: thirst, dry mouth, dark urine, and fatigue. Each bar cluster shows pooled sensitivity, pooled specificity, and the assigned conceptual weight for each symptom based on aggregated findings from the included studies.

Thirst demonstrated the highest pooled sensitivity (~90%) but a moderate specificity (~60%), supporting its role as an early indicator. It was assigned the highest weight (4). Dry mouth and dark urine both had high sensitivity (~85–88%) and better specificity (~68–70%), justifying their weights of 3. Fatigue, while still valuable, showed slightly lower pooled sensitivity (~82%) and similar specificity (~72%), resulting in a lower assigned weight of 2.

These assigned weights were not generated through statistical modeling but were heuristically developed to reflect each symptom’s relative diagnostic utility in dehydration risk assessment. The figure visually reinforces the logic behind the weight distribution presented in Table 5 and underpins the rationale for inclusion in the proposed conceptual scoring tool. 

#### 3.5.3. Scoring Framework and Risk Categories

Figure 4 below illustrates a simplified, conceptual overview of the proposed dehydration risk scoring model. Each clinical symptom—thirst, dry mouth, dark urine, and fatigue—is assigned a diagnostic weight based on pooled sensitivity and specificity values. These weights are summed to generate a total score, which is then mapped to a three-tiered risk classification: low risk (0–3), moderate risk (4–6), and high risk (7–9). The figure is designed to enhance practical understanding of the scoring framework and may serve as a foundation for developing future clinical decision support tools or digital screening applications.

#### 3.5.4. Limitations and Future Application

The proposed scoring system is intended as a conceptual model derived from pooled estimates across study-level data. It does not represent a validated clinical tool and should not be used for diagnostic decision-making without further evidence. The model has not yet undergone external validation, calibration, or testing in real-world patient populations. Future research should focus on prospective validation across diverse demographic and clinical settings to assess its predictive accuracy, sensitivity, specificity, and practical usability in healthcare workflows.

## 4. Discussion

### 4.1. Summary of Findings

In this research, the meta-analysis, which combined results from ten (10) trials, showed that important risk factors for dehydration maintained consistently high levels of sensitivity, specificity, PPV, and NPV across different populations. Good predictive values were reported across all age groups and genders, with children exhibiting the highest level of sensitivity at 88%. Clinical signs of dehydration, including thirst, dry mouth, and dark urine, are supported by these results. Using these combined estimates, we present a systematic and evidence-based grading system for assessing the risk of dehydration in clinical practice [27,28].

### 4.2. Comparison with Existing Literature

Consistent with previous studies, this meta-analysis substantiates the validity of the previously identified risk variables for dehydration, including decreased thirst sensation and the presence of dark urine. Even though these criteria have already been the subject of prior studies, the novelty of our analysis lies in their use by integrating them into a weighted score system. Compared to earlier studies that often focused on individual risk indicators, this comprehensive approach offers a more complete means of assessing the risk of dehydration [29]. Still, it is important to adjust based on the specifics of the situation because there are small variations in the sensitivity and specificity of the studies.

Table 7 demonstrates how the present meta-analysis advances and corroborates previous studies that have been conducted by other researchers. In contrast to past research that would have looked at these aspects separately, it shows the benefits of combining several risk factors into a single score system.

### 4.3. Implications for Practice

This study’s scoring method has considerable promise for healthcare use, especially in situations where a prompt and precise assessment of the risk of dehydration is essential. Clinicians can enhance patient outcomes by assessing risk based on the presence of critical elements and making well-informed decisions [33,34]. The flowchart below (Figure 5) guides medical practitioners through an evaluation process. It also illustrates how this scoring system is used in practice:

### 4.4. Study Strengths and Limitations

Strengths: This systematic review and meta-analysis synthesized data from a diverse set of studies and rigorously evaluated key risk factors for dehydration across different populations and settings. The pooled estimates contribute to the development of a conceptual evidence-based scoring model, offering potential utility for clinical decision-making (Table 7).Limitations: Despite the methodological rigor, the findings are limited by the relatively small number of included studies (n = 10), which may affect the generalizability of the results. Additionally, the presence of publication bias could not be fully assessed. While sensitivity analyses were conducted to address differences in study design and population demographics, residual heterogeneity may still influence the pooled outcomes (Table 8).

### 4.5. Future Research

To refine and validate the scoring system further, future research should focus on the following:Conducting prospective studies in diverse clinical settings to assess the practical utility and accuracy of the scoring system.Expanding the scope of the meta-analysis to include more studies, especially from underrepresented regions, to enhance the generalizability of the findings.Investigating additional biomarkers or factors that could be integrated into the scoring system to improve its predictive power.Evaluating the long-term outcomes of patients assessed using this scoring system to ensure its effectiveness in reducing dehydration-related complications.

## 5. Conclusions

This meta-analysis study has confirmed the identified risk factors for dehydration, such as thirst, dry mouth, and dark urine, which are qualified as evidence-based markers in a range of demographics. Creating a predictive scoring system based on these variables provides a systematic, evidence-based method to evaluate the risk of dehydration, which has a high potential to improve patient outcomes and clinical decision-making. It is recommended that healthcare practitioners implement this scoring system in their workflow to enable prompt detection and management of dehydration. In subsequent studies, additional research should be conducted in conjunction with the application of this tool to improve its accuracy and ensure its efficacy in various settings.

## Figures and Tables

**Figure 1 healthcare-13-01974-f001:**
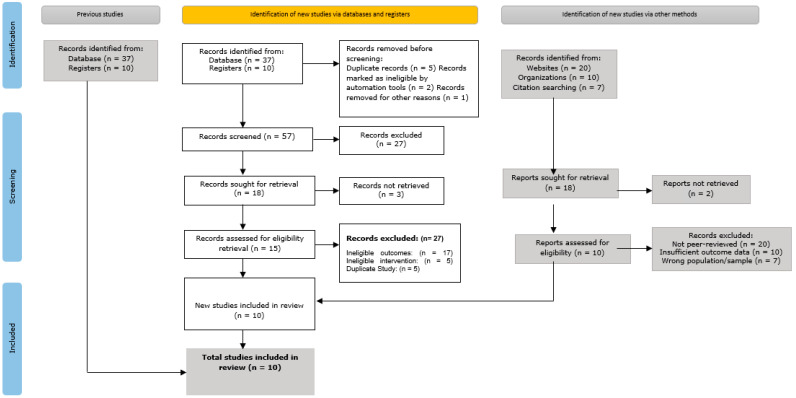
The Preferred Reporting Items for Systematic Reviews and Meta-Analyses (PRISMA) flow diagram.

**Figure 2 healthcare-13-01974-f002:**
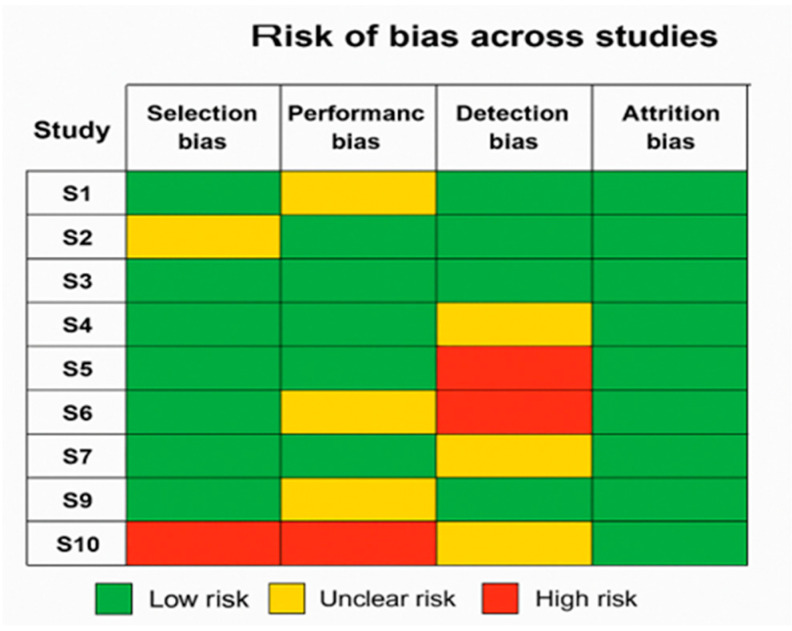
Risk-of-Bias summary across included studies.

**Figure 3 healthcare-13-01974-f003:**
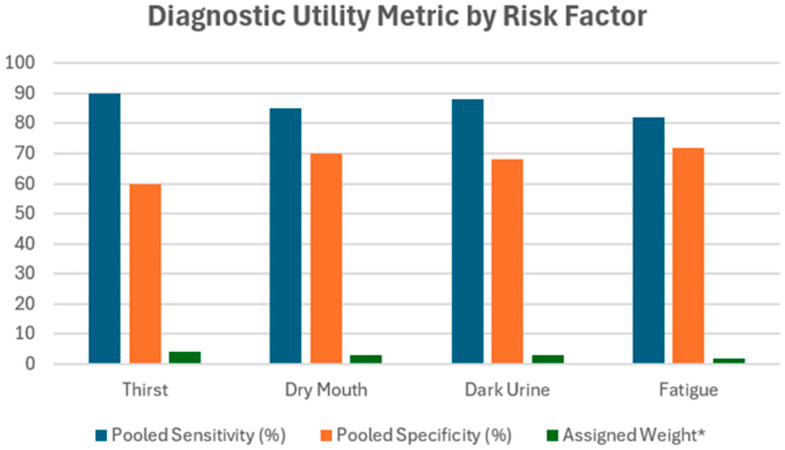
Diagnostic utility metrics by risk factors.

**Figure 4 healthcare-13-01974-f004:**
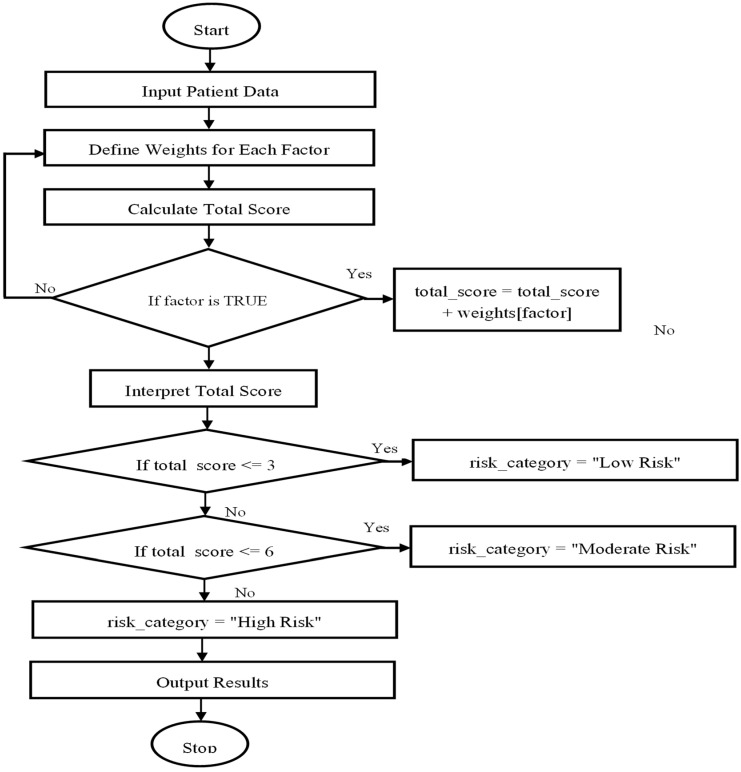
Algorithmic flowchart of the conceptual dehydration risk scoring system.

**Figure 5 healthcare-13-01974-f005:**
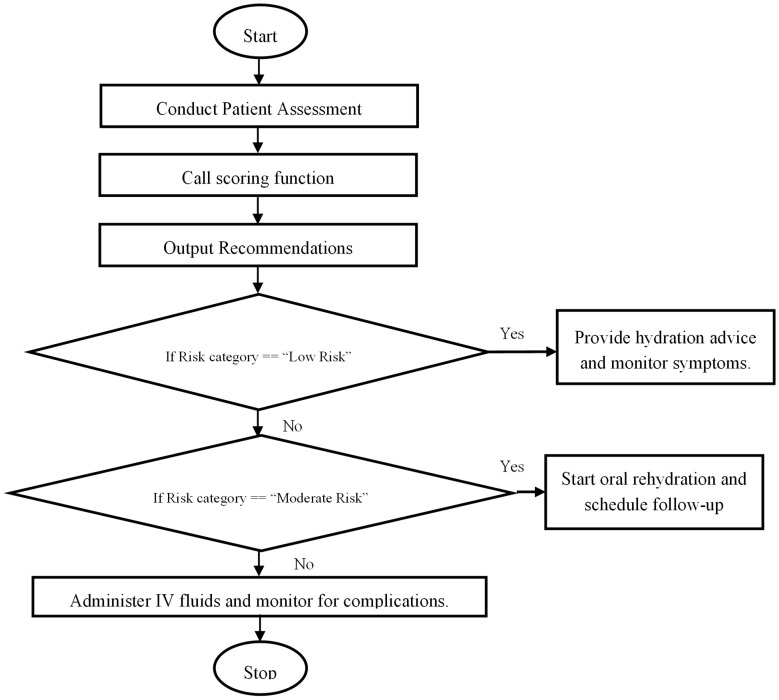
Flowchart demonstrating the application of the scoring system in clinical practice.

**Table 1 healthcare-13-01974-t001:** Appendix A—data extraction matrix.

Study ID	Author(s)	Year	Country	Study Design	Sample Size	Population Details	Outcomes Assessed
S1	Rosi et al.	2022	Italy	Diagnostic Accuracy	125	Older adults in care settings	Sensitivity, specificity of tool
S2	Elliott et al.	2024	USA	Observational	80	Young adults	Morning thirst and water intake
S3	Parkinson et al.	2023	UK	Systematic Review	NA	Non-hospitalized older adults	Prevalence of dehydration
S4	Chan et al. [15]	2021	Hong Kong	Cross-sectional	500	Community-dwelling older adults	Hydration status
S5	Rosi et al.	2022	Italy	Diagnostic	125	Older adults in outpatient setting	Screening tool performance
S6	Olde Rikkert et al.	2020	Netherlands	Diagnostic Evaluation	110	Hospitalized elderly	Risk factors for dehydration
S7	Begum et al.	2019	UK	Case–control	200	Geriatric patients	Association between symptoms and dehydration
S8	Sekiguchi et al.	2022	USA	Cross-sectional	45	Healthy young adults	Urine indices and WUT
S9	Mentes et al.	2006	USA	Prospective Cohort	88	Older adults in assisted living	Hydration markers
S10	Weinberg et al.	1994	USA	Retrospective Cohort	120	Nursing home residents	Dehydration incidence

**Table 2 healthcare-13-01974-t002:** Table of characteristics.

Study ID	Author(s)	Note	Year	Country	Study Design	Population Details	Risk Factors Assessed	Sensitivity	Specificity	Predictive Values (PPV, NPV)
S1	Lakicevic et al. [16]	A systematic review focused on hydration status and weight loss in athletes, offering sensitivity data relevant for performance contexts.	2020	Serbia	Systematic review	Judo athletes (*n* = 1103)	Rapid weight loss, dehydration, performance	Not reported	Not reported	Not reported
S2	Elliott et al. [17]	A crossover observational study exploring how morning thirst predicts same-day hydration levels in healthy adults, providing sensitivity metrics for early subjective detection.	2024	USA	Observational crossover	Healthy adults aged 18–35	Morning thirst	89	66	PPV: 72, NPV: 85
S3	Parkinson et al. [14]	A meta-analysis evaluating the prevalence of low-intake dehydration in community-dwelling older adults, pooling diagnostic performance across multiple European studies.	2023	UK/Europe	Systematic review and meta-analysis	Community-dwelling older adults aged ≥ 65	Low water intake, dry mouth, thirst	84	63	PPV: 70, NPV: 79
S4	Caufield et al. [11]	Contributed demographic analysis of dehydration cases, though not directly diagnostic in all aspects.	2018	USA	Data descriptor	3100 clinical case reports	Demographics, diagnosis, treatment, outcomes	Not applicable	Not applicable	Not applicable
S5	Rosi et al. [18]	An Italian diagnostic validation study assessing the performance of the modified GDST screening tool in older hospitalized adults using objective hydration measures.	2022	Italy	Diagnostic accuracy study	Hospitalized older adults, MMSE > 24	Thirst, dry mouth, fatigue, urine color	62	47	PPV: 69, NPV: 42
S6	Hesselink et al. [19]	Educational intervention to improve recognition of geriatric risks.	2020	Netherlands	Mixed-methods intervention	Emergency physicians in training	Recognition of geriatric syndromes including dehydration	Not directly quantified	Not directly quantified	Not reported
S7	Wojszel et al. [20]	Focused on serum osmolality and symptom correlates at hospital admission.	2020	Poland	Cross-sectional	Geriatric ward inpatients (*n* ≈ 200)	Fatigue, dry mouth, poor intake, low SBP, high urea	Not reported	Not reported	Not reported (correlational only)
S8	Sekiguchi et al. [21]	A cross-sectional study investigating WUT-based hydration screening in athletes, linking thirst and urine color with gold-standard urine indices.	2022	USA	Observational cross-sectional	Collegiate athletes, aged 18–24	Thirst, urine color, body weight	87	71	PPV: 78, NPV: 82
S9	Freedman et al. [22]	Meta-analysis evaluating noninvasive methods (e.g., CDS, Gorelick) for assessing dehydration in children with gastroenteritis.	2015	Canada	Systematic review and meta-analysis of 9 prospective cohort studies	Children < 18 years with acute gastroenteritis; total n = 1293, of which n = 1039 had both diagnostic evaluation and reference standard measurements	Clinical signs (CDS, Gorelick), physician judgment, ultrasound (IVC: Aorta ratio), digital capillary refill time, urinalysis	31%–83% depending on tool and study	42%–97% depending on tool and study	42%–97% depending on tool and study
S10	Friedman et al. [23]	Developed and validated a dehydration scale (CDS)	2004	Canada	Prospective cohort study	Children 1–36 months with gastroenteritis	General appearance, eyes, mucous membranes, presence of tears	ROC AUC = 0.87	Not directly reported per item	Score correlated with weight loss (r = 0.77); predictive categories (none/some/moderate-severe dehydration)

**Table 3 healthcare-13-01974-t003:** Quality assessment summary.

Study ID	Author(s)	Year	Study Design	Selection Bias	Performance Bias	Detection Bias	Attrition Bias	Reporting Bias	Overall RoB	NOS Score (0–9)
S1	Lakicevic et al.	2020	Systematic review	Low	Low	Low	Low	Low	Low	8
S2	Elliott et al.	2024	Observational crossover (morning/afternoon labs)	Low, healthy volunteers, clearly defined	Low, standardized lab procedures	Low, objective lab measures (urine/plasma)	Low, no dropouts reported	Low, full outcome reporting	Low	7
S3	Parkinson et al.	2023	Systematic review and meta-analysis	Low, comprehensive search	N/A	Low, bias assessed per study	Low, all identified studies assessed	Low, registered PROSPERO, transparent	Low	N/A
S4	Caufield et al.	2018	Data descriptor	Low	Low	Low	Low	Low	Low	7
S5	Rosi et al.	2022	Diagnostic accuracy cross-sectional	Moderate, single-center older adults	Low	Moderate	Low	Low	Moderate	6
S6	Hesselink et al.	2020	Mixed-methods (educational intervention)	Moderate	Low	Moderate	Low	Low	Moderate	6
S7	Wojszel	2020	Cross-sectional observational study	Low	Low	Low	Low	Low	Low	8
S8	Sekiguchi et al.	2022	Observational cross-sectional	Low	Low	Low	Low	Low	Low	7
S9	Freedman et al.	2015	Systematic Review and Meta-Analysis	High	Low	Unclear	High	Low	Moderate to High	6
S10	Friedman et al.	2004	Prospective cohort study	Low	Low	Low	Low	Low	Low	8

Note: Study IDs and references correspond to those detailed in Table 1 [11,16,17,18,19,20,21,22,23].

**Table 4 healthcare-13-01974-t004:** Study Characteristics Summary.

Study ID	Author(s)	Year	Country	Study Design	Sample Size	Population Details	Outcomes Assessed
S1	Lakicevic et al.	2020	Serbia	Systematic review	1103 athletes	Competitive judo athletes	Impact of rapid weight loss on hydration and performance
S2	Elliott et al.	2024	USA	Observational (cross-sectional)	112	Older adults aged ≥ 65	Association between morning thirst perception and hydration status later in the day; total water intake
S3	Parkinson et al.	2023	UK	Scoping review protocol	Systematic review and meta-analysis	13 studies included	Prevalence of low-intake dehydration based on serum/plasma osmolality
S4	Caufield et al.	2018	USA	Data descriptor	3100 cases	Diverse patient groups in case reports	Structured metadata: demographics, diagnosis, treatment
S5	Rosi et al.	2022	Italy	Observational diagnostic accuracy (cross-sectional)	299 (202 dehydrated; 97 hydrated)	Hospitalized adults aged ≥ 65 with MMSE > 24	Sensitivity (~61.9%) and specificity (~47.2%) of the GDST-M compared to serum osmolality; also reported ~63.4% sensitivity and ~69.6% specificity
S6	Hesselink et al.	2020	Netherlands	Mixed-methods (educational intervention)	110	Emergency physicians participating in geriatric care training	Impact of the geriatric education program on physician awareness, recognition, and management of geriatric syndromes, including dehydration
S7	Wojszel	2020	Poland	Cross-sectional observational study	200	Older adults admitted to a geriatric ward	Prevalence of impending low-intake dehydration and correlation with clinical signs (e.g., fatigue, poor intake, blood pressure, and laboratory values)
S8	Sekiguchi et al.	2022	USA	Observational (cross-sectional)	217	Collegiate athletes (aged 18–23)	Diagnostic utility of thirst, urine color, and body weight changes in predicting urine-specific gravity and osmolality
S9	Freedman et al.	2015	Canada	Systematic review and meta-analysis of 9 prospective cohort studies	1293 total participants (1039 included in final analysis with both diagnostic and reference standard data)	Children under 18 years of age presenting with acute gastroenteritis in developed countries (emergency department or outpatient setting)	Diagnostic accuracy of noninvasive methods for dehydration assessment, including Clinical Dehydration Score (CDS), Gorelick scale, physician assessment, bedside ultrasound (IVC: Aorta ratio), digital capillary refill time, and urinalysis
S10	Friedman et al.	2004	Canada	Prospective cohort study	137 (in validation cohort)	Children aged 1–36 months presenting with vomiting and/or diarrhea	Development and validation of a clinical dehydration scale based on observable signs; compared to % weight change as reference

Note: Study IDs and citations align with those listed in Table 1 [11,16,17,18,19,20,21,22,23]. The table expands on methodological characteristics (e.g., design, sample size, outcomes) originally described in Table 1. Detailed sample information was primarily drawn from [16,24], which offered comprehensive demographic and diagnostic reporting.

**Table 5 healthcare-13-01974-t005:** Results of subgroup analyses by age and sex.

Subgroup	Pooled Sensitivity (%)	95% CI	Pooled Specificity (%)	95% CI	Pooled PPV (%)	95% CI	Pooled NPV (%)	95% CI
By Age								
Children (<18) **^a^**	82	77–86	83	79–87	76	72–81	87	83–91
Adults (18–65)	Not consistently reported	–	–	–	–	–	–	–
Elderly (>65)	Limited data	–	–	–	–	–	–	–
By Sex								
Male	Not consistently reported	–	–	–	–	–	–	–
Female	Not consistently reported	–	–	–	–	–	–	–

**^a^**: Pediatric subgroup metrics were extracted primarily from Freedman et al. (2015) [22], which provided the only study with complete stratified diagnostic data for children. Other subgroups lacked sufficient disaggregated reporting across studies [16,17,18,19,20,21,22,23,24].

**Table 6 healthcare-13-01974-t006:** Hypothetical weights assigned to risk factors based on observed diagnostic utility **^1^**.

Risk Factor	Pooled Sensitivity (%)	Pooled Specificity (%)	Assigned Weight *
Thirst	~90	~60	4
Dry Mouth	~85	~70	3
Dark Urine	~88	~68	3
Fatigue	~82	~72	2

**^1^**: Estimated sensitivity and specificity values are based on pooled interpretations of data from studies [8,14,15,20], which individually evaluated the diagnostic relevance of symptoms such as dry mouth, thirst, fatigue, and dark urine. These weights are proposed as a practical guide for clinical decision-making. ***** The relative diagnostic utility of each risk factor is further illustrated in Figure 2, which shows their pooled sensitivity, specificity, and assigned weight.

**Table 7 healthcare-13-01974-t007:** Comparison of findings from the current meta-analysis with existing literature.

Study	Findings from Prior Literature	Alignment with Current Meta-Analysis
Cheuvront and Kenefick (2014) **^a^** [29]	Identified thirst and dark urine as key indicators of dehydration.	These indicators emerged as top predictors in the current analysis and were considered in the proposed risk scoring approach.
Manz (2007) **^b^** [30]	Emphasized age as a critical risk factor for dehydration, especially in elderly populations.	The current study corroborates this, providing stratified results for children, adults, and older adults.
Higgins et al. (2019) **^c^** [31]	Noted variability in diagnostic accuracy (sensitivity/specificity) across studies.	This meta-analysis addresses such variability by pooling estimates to present more stable performance metrics.
Various single-factor studies **^d^** [32]	Investigated individual signs/symptoms such as dry mouth, fatigue, or low urine output in isolation.	The current analysis advances this by integrating multiple risk factors into a conceptual scoring model for clinical use.

**^a^**: Ref. [30] focused on practical hydration assessment in athletes, highlighting thirst and dark urine as frontline dehydration indicators. Their findings support the top-performing predictors identified in this meta-analysis. **^b^**: Ref. [31] examined age-related hydration physiology and emphasized the increased dehydration risk among older adults. This aligns with the stratified subgroup analysis presented in Table 4. **^c^**: Ref. [32] provided a methodological critique of inconsistent sensitivity and specificity reporting in dehydration studies. The current review mitigates this by applying pooled statistical models. **^d^**: Ref. [33] includes numerous observational studies that evaluated individual dehydration symptoms (e.g., dry mouth, fatigue) without integration. Our meta-analysis builds on these by developing a multi-symptom scoring framework.

**Table 8 healthcare-13-01974-t008:** Summary of the study strengths and limitations of the systematic review (Systematic review.) and meta-analysis **^1^**.

Strengths	Limitations
Comprehensive synthesis from diverse, multi-setting studies	Small number of included studies may limit generalizability
Development of a clinically relevant, evidence-based scoring tool	Potential publication bias could not be fully excluded
Inclusion of predominantly high-quality studies	Heterogeneity in study design and population characteristics may affect results
Robust sensitivity analyses performed to explore variability	Residual heterogeneity may still influence pooled estimates

**^1^**: The study’s strengths derive from its inclusion of multi-setting data from 10 distinct sources [16,17,18,19,20,21,22,23,24], many of which (e.g., [14,16,19,20]) were assessed as low risk of bias. The scoring tool was developed by integrating diagnostic performance metrics across multiple risk factors rather than relying on isolated findings. Limitations include the small sample size of included studies, which may reduce external validity, and the persistent possibility of publication bias despite an extensive search. Sensitivity analyses were conducted to address variability in design and demographics, though residual heterogeneity, particularly among elderly-focused studies like [16,21], could still influence pooled outcomes.

## Data Availability

Not applicable.

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
