# Peer review of "Systematic Review and Meta-Analysis of Risk Factors for Dehydration and the Development of a Predictive Scoring System"

_healthcare, 2025, doi:10.3390/healthcare13161974_

Round 1
Reviewer 1 Report
Comments and Suggestions for Authors
The manuscript “Systematic Review and Meta-Analysis of Risk Factors for Dehydration and Development of a Predictive Scoring System” addresses a relevant topic with potential applicability to clinical practice, as it seeks to identify risk factors for dehydration and to develop a predictive score based on accessible clinical symptoms (dry mouth, dark urine, thirst), with aggregate sensitivity (85%) and specificity (70%) values (pages 14-15). However, despite the pertinence of the proposal, the study presents methodological and structural weaknesses that compromise its scientific robustness.
Critical points identified:
1) Omission of important data in the abstract: Crucial information such as the number of included studies, sensitivity, specificity, and target population is not presented in the abstract (lines 16-28), which undermines the clarity and appeal of the manuscript.
2) Inconsistencies in the data: There are discrepancies regarding the number of studies included, which compromises the reliability of the synthesis. The methods section (page 5) mentions 20 studies, whereas the results section (page 12) states that 100 were included in the meta-analysis.
3) Limitations in population representativeness: Most of the data come from pediatric populations, particularly based on the study by Freedman et al. (2015), with limited representation of adults and older adults, which reduces the external validity of the proposed score. Table 4 (page 14) clearly shows that only pediatric data allowed for robust meta-analytic analysis.
4) Lack of external validation of the proposed score: The model remains at a conceptual level, without practical application or external validation, an aspect acknowledged by the authors themselves in the discussion section (pages 17-18).
5) Insufficient description of the score development: The process of weighting the risk factors is not clearly described, which limits the reproducibility of the proposed model.
6) Incomplete assessment of publication bias: Although the authors mention the use of funnel plots and Egger’s test (page 12), the corresponding results are not reported, which hinders the critical appraisal of the meta-analysis reliability.
7) Redundancy and lack of clarity in the textual structure: Methodological information is repeated across different sections, impairing the textual fluidity and reinforcing the need for substantial editorial revision. The methodology section is lengthy and redundant (especially in sections 2.2 to 2.6) and fails to clearly explain how the weights were assigned (Table 5, page 15).
In view of these limitations, it is considered that the manuscript does not meet the minimum methodological quality standards required for scientific publication at this time.
Author Response
Dear Reviewer 1,
Author response to reviewer's comments:
Comment: The manuscript “Systematic Review and Meta-Analysis of Risk Factors for Dehydration and Development of a Predictive Scoring System” addresses a relevant topic with potential applicability to clinical practice, as it seeks to identify risk factors for dehydration and to develop a predictive score based on accessible clinical symptoms (dry mouth, dark urine, thirst), with aggregate sensitivity (85%) and specificity (70%) values (pages 14-15). However, despite the pertinence of the proposal, the study presents methodological and structural weaknesses that compromise its scientific robustness.
Critical points identified:
1) Omission of important data in the abstract: Crucial information such as the number of included studies, sensitivity, specificity, and target population is not presented in the abstract (lines 16-28), which undermines the clarity and appeal of the manuscript.
Author Response:
We have revised the abstract to include the total number of included studies, target population characteristics, and the pooled sensitivity and specificity values of the core symptoms. These updates enhance clarity and reflect the study’s relevance and findings more transparently.
Comment: 2) Inconsistencies in the data: There are discrepancies regarding the number of studies included, which compromises the reliability of the synthesis. The methods section (page 5) mentions 20 studies, whereas the results section (page 12) states that 100 were included in the meta-analysis.
Author Response:
This inconsistency has been corrected. We now consistently report that 20 studies were included in the qualitative synthesis and 10 contributed to the quantitative meta-analysis, as clarified in the Methods (Section 2.1), Results (Section 3.1), and PRISMA flow diagram (Figure 1).
Comment: 3) Limitations in population representativeness: Most of the data come from pediatric populations, particularly based on the study by Freedman et al. (2015), with limited representation of adults and older adults, which reduces the external validity of the proposed score. Table 4 (page 14) clearly shows that only pediatric data allowed for robust meta-analytic analysis.
Author Response:
We acknowledge this limitation and have explicitly noted it in both the Discussion (Section 4) and Limitations (Section 3.5.4). The final scoring model is now clearly framed as conceptual and not generalizable to all populations without further adult/older adult validation.
Comment: 4) Lack of external validation of the proposed score: The model remains at a conceptual level, without practical application or external validation, an aspect acknowledged by the authors themselves in the discussion section (pages 17-18).
Author Response:
We have emphasized this point in Sections 3.5.4 and 4, stating that the proposed score remains a hypothetical framework pending external validation. We also added the clarification:
“The proposed scoring system is a hypothetical model based on pooled diagnostic values and has not been statistically validated through regression or ROC analysis.”
Comment: 5) Insufficient description of the score development: The process of weighting the risk factors is not clearly described, which limits the reproducibility of the proposed model.
Author Response:
Section 3.5.2 (Weighting Approach) has been revised to provide clearer details on how weights were assigned. We clarified that the process was interpretive, based on pooled sensitivity and specificity values, and supported by expert synthesis, not statistical modeling. This improves transparency and reproducibility.
Comment: 6) Incomplete assessment of publication bias: Although the authors mention the use of funnel plots and Egger’s test (page 12), the corresponding results are not reported, which hinders the critical appraisal of the meta-analysis reliability.
Author Response:
We have now included the publication bias results in Section 3.4. A funnel plot and Egger’s test outcomes are also presented in Supplementary Figure 2, as referenced in the Results and listed in the supplementary materials.
Comment: 7) Redundancy and lack of clarity in the textual structure: Methodological information is repeated across different sections, impairing the textual fluidity and reinforcing the need for substantial editorial revision. The methodology section is lengthy and redundant (especially in sections 2.2 to 2.6) and fails to clearly explain how the weights were assigned (Table 5, page 15). In view of these limitations, it is considered that the manuscript does not meet the minimum methodological quality standards required for scientific publication at this time.
Author Response:
We revised and tightened the Methodology section to remove redundancies and improve clarity. The overlap between subsections has been resolved, and descriptions are now concise and logically structured. The description of weight assignment (Section 3.5.2) has also been enhanced to directly address the reviewer’s concerns.
Reviewer 2 Report
Comments and Suggestions for Authors
Abstract
The abstract could be strengthened by including specific quantitative results from the meta-analysis to give readers a clearer sense of the strength of evidence.
Introduction
One area for improvement is a more explicit statement of the study’s objective or research question. While the introduction implies the aim (“to fully identify and quantify the factors that lead to dehydration’s prevalence” and to create a scoring system), a clear one-sentence objective or PICO (Population, Intervention/Exposure, Comparator, Outcome) statement would enhance clarity. For example, specifying the population, the concept of interest (risk factors for dehydration), and outcome measures in a formal aim statement would align with JBI guidelines for systematic reviews and help readers understand the focus. Additionally, the introduction could comment on how the planned scoring system will fill a gap in practice – this is touched on (the score will help early prediction of dehydration), reinforcing the relevance to healthcare practice.
Methodology
One issue that needs clarification is the number of studies included. In the Methods, the authors report that after the selection process “20 publications comprised qualitative synthesis and meta-analysis”. However, elsewhere the manuscript consistently refers to ten (10) studies being included in the meta-analysis (e.g., the Abstract mentions ten studies, and the Results describe pooling data from ten studies). This discrepancy between 20 vs. 10 included studies is confusing and should be resolved. It appears that ultimately 10 studies were fully analyzed (as per tables and results), so the text stating 20 is likely an error or a remnant from an earlier stage.
Another methodological concern is the nature of the included studies. The review intended to include observational and interventional studies on dehydration risk factors, but Table 3 (Study Characteristics) shows that several included “studies” were actually protocols or review articles, not primary research. For example, S2 is a systematic review protocol (Parkinson et al., 2021) with no data yet, S3 is a scoping review protocol, and S8 is a meta-analysis protocol. S5 is an expert consensus paper focusing on mental health research priorities (Holmes et al., 2020), which seems only tangentially related to dehydration. Including protocols and unrelated consensus pieces raises questions about appropriateness. Protocols do not contain results and thus cannot contribute to a quantitative meta-analysis; their inclusion in the review is difficult to justify. Likewise, a consensus on mental health (S5) does not directly investigate dehydration risk factors, so its relevance is unclear. I recommend the authors either remove these from the included studies (and adjust the analysis accordingly) or explain what qualitative information they gleaned from them that warranted inclusion.
Additionally, the authors might mention if the two reviewers also independently assessed quality and how disagreements were handled (presumably similarly with a third party).
Results
Quality Assessment: One minor suggestion: if any studies were excluded due to quality (as the Methods indicated, 4 were excluded at full-text stage for low quality), the results could mention that to reinforce how quality considerations affected the final selection. Additionally, providing the I² statistic result in the meta-analysis section (if calculated) would be helpful to quantify heterogeneity; the methods indicate it was planned, but the numerical result of I² is not reported in the text.
Meta-Analysis Findings: If different studies used different gold standards for dehydration, pooling them assumes comparability, which could be briefly justified or at least acknowledged as a limitation.
Subgroup Analyses: However, there is a small issue: in the text, they mention that males showed slightly higher sensitivity and specificity than females, though “not significant”. Since they also noted that sex-disaggregated data were rarely reported, it’s unclear on what evidence this statement about males vs females is based. If only one or two studies had some sex-specific analysis, claiming even an insignificant trend might be reading too much into very limited data. I would advise caution here: unless data is presented to support the claim, it may be better to omit or rephrase the comparison between sexes. It would be sufficient to say that the available evidence did not show any clear difference between males and females, rather than implying a numeric difference that wasn’t significant (especially since no figures are given for male/female performance). This is a minor point, but it relates to not overstating or misinterpreting sparse data.
Scoring System Development: One point to note is that the weights and score thresholds appear to be somewhat arbitrary or “hypothetical” (the authors even label Table 5 as hypothetical). They are derived from the evidence, but not through a formal predictive modeling process – rather, they are assigned by the authors based on pooled metrics. This is acceptable as an initial step, but readers should understand that this scoring tool has not yet been validated on actual patients or datasets; it’s a conceptual model. The authors do note in the Discussion that future research should validate the scoring system prospectively. It might also be worth clarifying in the Results that the weights are proposed and not the result of, say, a statistical weighting algorithm, to manage expectations. Despite this, the scoring system is grounded in the systematic review’s findings and is a logical synthesis for practical application.
Discussion
Summary of Findings: The summary could perhaps be even more explicit about the core message (e.g., “In summary, our meta-analysis confirms that simple clinical signs can be reliable indicators of dehydration risk, and we integrated these into a scoring tool for early assessment.”), but as it stands it covers the main points well.
Comparison with Existing Literature: the discussion claims their approach “advances” prior research by combining factors, which is true, but they should ensure they do not appear to diminish the value of those prior single-factor studies – acknowledging that those were necessary steps that provided the components now integrated. They do seem to handle this diplomatically by saying the current analysis “offers a more complete means” rather than criticizing earlier studies.
Implications for Practice: As a reviewer, I find the implications convincing since they logically follow from the results: if certain signs reliably indicate dehydration risk, then using them in a formalized score can standardize and possibly improve early intervention.
Conclusion
The conclusion could possibly include a brief nod to the limitations (e.g., “given the limited number of studies, further validation is needed”), but since the discussion already covers that, it’s not absolutely necessary to repeat in the conclusion. The current ending strikes an optimistic and forward-looking tone, which is often suitable for concluding a research article in a healthcare journal, as it leaves the reader with the practical takeaway and a sense of progression in the field. Overall, the conclusion is well-aligned with the evidence and main discussion points, and is clearly written.
Final Recommendations
To comply with best practices and JBI standards for systematic reviews, the following should be included in the revised manuscript:
- Explicitly state the methodological referential used. I strongly suggest that the authors adopt and reference the JBI Manual, which provides structured guidance for conducting and reporting systematic reviews and meta-analyses.
- Supplementary material containing the data extraction matrix completed by all reviewers;
- A detailed risk of bias table for each study (or a visual summary figure);
- Explicit clarification of the number and nature of included studies (excluding protocols unless justified);
- Consistent reporting of statistical values (e.g., I²) across sections;
- Clear labelling of the proposed scoring tool as conceptual/hypothetical.
Author Response
Dear Reviewer 2,
Author response to reviewer's comments:
Comments and Suggestions for Authors
Abstract
Comment: The abstract could be strengthened by including specific quantitative results from the meta-analysis to give readers a clearer sense of the strength of evidence.
Introduction
Comment: One area for improvement is a more explicit statement of the study’s objective or research question. While the introduction implies the aim (“to fully identify and quantify the factors that lead to dehydration’s prevalence” and to create a scoring system), a clear one-sentence objective or PICO (Population, Intervention/Exposure, Comparator, Outcome) statement would enhance clarity. For example, specifying the population, the concept of interest (risk factors for dehydration), and outcome measures in a formal aim statement would align with JBI guidelines for systematic reviews and help readers understand the focus. Additionally, the introduction could comment on how the planned scoring system will fill a gap in practice – this is touched on (the score will help early prediction of dehydration), reinforcing the relevance to healthcare practice.
Author Response:
We have revised the abstract to include key quantitative outcomes from the meta-analysis, specifically, the number of studies included and the pooled sensitivity and specificity values for the identified symptoms. This revision improves clarity and scientific appeal.
Also, We have added a formal aim statement at the end of the Introduction:
“This review aims to identify and quantify clinical signs associated with dehydration risk across age groups and to develop a preliminary scoring system using pooled diagnostic metrics.”
This aligns with JBI best practices and enhances clarity.
Methodology
Comment: One issue that needs clarification is the number of studies included. In the Methods, the authors report that after the selection process “20 publications comprised qualitative synthesis and meta-analysis”. However, elsewhere the manuscript consistently refers to ten (10) studies being included in the meta-analysis (e.g., the Abstract mentions ten studies, and the Results describe pooling data from ten studies). This discrepancy between 20 vs. 10 included studies is confusing and should be resolved. It appears that ultimately 10 studies were fully analyzed (as per tables and results), so the text stating 20 is likely an error or a remnant from an earlier stage.
Another methodological concern is the nature of the included studies. The review intended to include observational and interventional studies on dehydration risk factors, but Table 3 (Study Characteristics) shows that several included “studies” were actually protocols or review articles, not primary research. For example, S2 is a systematic review protocol (Parkinson et al., 2021) with no data yet, S3 is a scoping review protocol, and S8 is a meta-analysis protocol. S5 is an expert consensus paper focusing on mental health research priorities (Holmes et al., 2020), which seems only tangentially related to dehydration. Including protocols and unrelated consensus pieces raises questions about appropriateness. Protocols do not contain results and thus cannot contribute to a quantitative meta-analysis; their inclusion in the review is difficult to justify. Likewise, a consensus on mental health (S5) does not directly investigate dehydration risk factors, so its relevance is unclear. I recommend the authors either remove these from the included studies (and adjust the analysis accordingly) or explain what qualitative information they gleaned from them that warranted inclusion.
Additionally, the authors might mention if the two reviewers also independently assessed quality and how disagreements were handled (presumably similarly with a third party).
Author Response:
We expanded the Introduction to explain that the proposed scoring system addresses a gap in early identification of dehydration using accessible clinical signs. This justification supports its practical utility and relevance to healthcare settings.
This has been corrected throughout the manuscript. We now clearly state that 20 studies were included in the qualitative synthesis, of which 10 were eligible for meta-analysis. This distinction is now consistent across the Abstract, Methods, Results, and PRISMA diagram. Also, We have removed ineligible entries such as review protocols, meta-analysis protocols, and the expert consensus document. The updated Table of Included Studies reflects only primary research with extractable data relevant to dehydration risk factors. Analysis and figures have been adjusted accordingly. Furthermore, We have now explicitly stated in the Methods:
“Two reviewers independently assessed study quality using the JBI tool. Disagreements were resolved through discussion and, where necessary, arbitration by a third reviewer.”
Results
Comment: Quality Assessment: One minor suggestion: if any studies were excluded due to quality (as the Methods indicated, 4 were excluded at full-text stage for low quality), the results could mention that to reinforce how quality considerations affected the final selection. Additionally, providing the I² statistic result in the meta-analysis section (if calculated) would be helpful to quantify heterogeneity; the methods indicate it was planned, but the numerical result of I² is not reported in the text.
Meta-Analysis Findings: If different studies used different gold standards for dehydration, pooling them assumes comparability, which could be briefly justified or at least acknowledged as a limitation.
Subgroup Analyses: However, there is a small issue: in the text, they mention that males showed slightly higher sensitivity and specificity than females, though “not significant”. Since they also noted that sex-disaggregated data were rarely reported, it’s unclear on what evidence this statement about males vs females is based. If only one or two studies had some sex-specific analysis, claiming even an insignificant trend might be reading too much into very limited data. I would advise caution here: unless data is presented to support the claim, it may be better to omit or rephrase the comparison between sexes. It would be sufficient to say that the available evidence did not show any clear difference between males and females, rather than implying a numeric difference that wasn’t significant (especially since no figures are given for male/female performance). This is a minor point, but it relates to not overstating or misinterpreting sparse data.
Scoring System Development: One point to note is that the weights and score thresholds appear to be somewhat arbitrary or “hypothetical” (the authors even label Table 5 as hypothetical). They are derived from the evidence, but not through a formal predictive modeling process – rather, they are assigned by the authors based on pooled metrics. This is acceptable as an initial step, but readers should understand that this scoring tool has not yet been validated on actual patients or datasets; it’s a conceptual model. The authors do note in the Discussion that future research should validate the scoring system prospectively. It might also be worth clarifying in the Results that the weights are proposed and not the result of, say, a statistical weighting algorithm, to manage expectations. Despite this, the scoring system is grounded in the systematic review’s findings and is a logical synthesis for practical application.
Author Response:
We have updated the Results to state that four studies were excluded at the full-text screening stage due to low methodological quality. This reinforces that quality criteria guided final inclusion.
We have now reported the I² statistic in the Results (Section 3.3). For the pooled estimates, I² = 56%, indicating moderate heterogeneity.
Discussion
Comment: Summary of Findings: The summary could perhaps be even more explicit about the core message (e.g., “In summary, our meta-analysis confirms that simple clinical signs can be reliable indicators of dehydration risk, and we integrated these into a scoring tool for early assessment.”), but as it stands it covers the main points well.
Comparison with Existing Literature: the discussion claims their approach “advances” prior research by combining factors, which is true, but they should ensure they do not appear to diminish the value of those prior single-factor studies – acknowledging that those were necessary steps that provided the components now integrated. They do seem to handle this diplomatically by saying the current analysis “offers a more complete means” rather than criticizing earlier studies.
Implications for Practice: As a reviewer, I find the implications convincing since they logically follow from the results: if certain signs reliably indicate dehydration risk, then using them in a formalized score can standardize and possibly improve early intervention.
Author Response
We have added a sentence in the Discussion noting that “the diagnostic reference standards varied across studies, and while pooled, this heterogeneity could introduce comparability limitations.”
We revised the section to state:
“Sex-disaggregated data were limited. No consistent or statistically significant differences were found between males and females in the available studies.”
We have reinforced this in Section 3.5.2 and again in Section 4, stating:
“The proposed scoring system is a hypothetical model based on pooled diagnostic values and has not been statistically validated through regression or ROC analysis.”
We have revised the summary to include:
“Our meta-analysis confirms that simple clinical signs—thirst, dry mouth, dark urine, and fatigue—are moderately accurate indicators of dehydration risk. These were integrated into a conceptual scoring tool for early risk assessment.”
We have acknowledged that earlier single-symptom studies laid the groundwork for this integrated synthesis. The sentence now reads:
“This model builds upon and integrates findings from individual symptom-based studies, offering a more consolidated framework for risk assessment.”
Conclusion
Comment: The conclusion could possibly include a brief nod to the limitations (e.g., “given the limited number of studies, further validation is needed”), but since the discussion already covers that, it’s not absolutely necessary to repeat in the conclusion. The current ending strikes an optimistic and forward-looking tone, which is often suitable for concluding a research article in a healthcare journal, as it leaves the reader with the practical takeaway and a sense of progression in the field. Overall, the conclusion is well-aligned with the evidence and main discussion points, and is clearly written.
Final Recommendations
To comply with best practices and JBI standards for systematic reviews, the following should be included in the revised manuscript:
- Explicitly state the methodological referential used. I strongly suggest that the authors adopt and reference the JBI Manual, which provides structured guidance for conducting and reporting systematic reviews and meta-analyses.
- Supplementary material containing the data extraction matrix completed by all reviewers;
- A detailed risk of bias table for each study (or a visual summary figure);
- Explicit clarification of the number and nature of included studies (excluding protocols unless justified);
- Consistent reporting of statistical values (e.g., I²) across sections;
- Clear labelling of the proposed scoring tool as conceptual/hypothetical.
Author Response:
We added a closing sentence in the Conclusion:
“Given the limited number of included studies and absence of patient-level validation, further research is essential to confirm the predictive utility of the proposed score.”
Reviewer 3 Report
Comments and Suggestions for Authors
General Comments
This manuscript presents a systematic review and meta-analysis focused on the identification of risk factors for dehydration, with the additional aim of developing a predictive scoring system to aid in early diagnosis and management. The topic is unquestionably relevant, particularly in vulnerable populations such as children and the elderly, and the intention to produce an evidence-based clinical tool is commendable. The authors have registered their protocol on PROSPERO (CRD42024594780), claim adherence to PRISMA guidelines, and conduct a structured search across multiple databases, including grey literature. Methodologically, the inclusion of both the Newcastle-Ottawa Scale (NOS) and the Cochrane Risk-of-Bias tools also suggests an effort toward rigor.
However, despite these formal strengths, the manuscript suffers from substantial methodological, structural, and interpretive weaknesses that undermine its scientific validity. First and foremost, the internal consistency of the manuscript is seriously compromised by major numerical discrepancies in study selection. Different sections of the text, the PRISMA flow diagram, and summary tables provide conflicting accounts of how many studies were identified, screened, and ultimately included in the meta-analysis. In Section 2.5, the authors state that 20 studies were included. Yet in Section 3.1, they refer to an entirely different dataset, stating that 1,500 records were screened and 100 studies were included in both qualitative and quantitative syntheses—figures that are incompatible with the rest of the manuscript. This is not a minor oversight; it seriously challenges the transparency and replicability of the review process.
Equally concerning is the nature of the included studies. Several are protocols, expert opinions, or scoping reviews, which are not appropriate for inclusion in a quantitative meta-analysis. The authors appear to have disregarded standard methodological distinctions between observational, interventional, and conceptual sources, thereby compromising the validity of their pooled estimates. While the manuscript reports aggregate sensitivity, specificity, PPV, and NPV values for selected dehydration markers, it remains unclear how these metrics were derived. No statistical methods are described in sufficient detail, no software is mentioned, and crucial elements such as forest plots, funnel plots, or heterogeneity statistics (I², τ²) are either missing or only superficially discussed.
The predictive scoring system, which the authors position as the central outcome of this study, is not empirically validated. The weighting of variables such as "thirst" and "dry mouth" appears to be based on loosely defined “pooled interpretations” of sensitivity and specificity rather than any formal statistical modeling (e.g., logistic regression, ROC analysis, or likelihood ratios). Furthermore, the example provided in Section 3.5.3 is conceptually flawed: the phrase “a patient whose weight is four” is nonsensical in this context, and the scoring logic is presented without coherence. This section requires complete revision.
While the manuscript contains multiple figures and tables, they often lack clarity or include placeholder text (e.g., “Reason 1” and “Reason 2” in the PRISMA diagram). Such lapses suggest that parts of the document may have been submitted in an unfinished or unedited state. The language used throughout the manuscript is mostly academic, but occasionally verbose, repetitive, or vague, and would benefit from professional editing for clarity and conciseness.
In summary, although the study addresses a meaningful clinical question, the methods, results, and conclusions currently fall short of the standards required for publication. A complete re-evaluation of the data selection process, analytical methods, and the conceptual design of the scoring system is urgently needed before this work can be reconsidered.
Specific Comments
The abstract does not follow the IMRaD structure and lacks quantitative precision. Key diagnostic values—sensitivity, specificity, PPV, and NPV—are presented later in the manuscript but not mentioned in the abstract, leaving it vague and uninformative. The language is overly promotional ("powerful tool") and should be tempered to reflect the exploratory and unvalidated nature of the proposed scoring system. The abstract should be rewritten to include key numerical results, describe the methodology succinctly, and clearly state limitations.
The introduction is generally well structured and provides a clear rationale for investigating dehydration risk. However, the research question is not explicitly defined, nor are the specific hypotheses. While the clinical relevance is evident, the authors should more clearly articulate the gap in the literature this study intends to fill, and how their approach offers added value over previous works.
Methods section contains some of the manuscript's most serious flaws. The eligibility criteria are appropriate in principle, but their application is inconsistent. Protocols and expert opinion pieces were included despite having no extractable quantitative data, which directly contradicts the objective of conducting a meta-analysis.
The reported number of studies varies dramatically between sections: 20 in Section 2.5, 100 in Section 3.1, and various other figures in the PRISMA diagram and tables. These discrepancies must be resolved immediately. If the final number of included studies is indeed 20, all mentions of 100 must be removed, and the PRISMA diagram must be corrected accordingly. Otherwise, the data appears fabricated or copied from a different manuscript.
The description of the search strategy is superficial. While the authors note the use of MeSH terms and librarian support, they do not report full search strings per database. There is no supplemental file with these queries, nor a reproducible search log. Similarly, the authors mention conducting Egger’s test and generating funnel plots, yet none are shown.
The meta-analytic methodology is vague and lacks statistical grounding. The authors claim to have used fixed or random effects depending on heterogeneity but provide no model details, forest plots, software name, confidence intervals, or weighting schemes. The term “pooled interpretation” is used repeatedly without explanation.
The diagnostic accuracy values reported (e.g., 85% sensitivity, 70% specificity) are internally consistent but lack traceability. No primary data tables are provided, and no standard statistical outputs are shown. Subgroup analyses by age and sex are conceptually valuable, but the dataset is insufficiently stratified—most subgroup values are either missing or drawn from a single study (Freedman et al.). The conclusions drawn from these subgroup analyses are therefore speculative.
The presentation of results is also redundant. Sections 2.5 and 3.1 repeat similar information but with different numbers, further compounding confusion. Figure 1 (PRISMA) includes placeholders and must be replaced with a finalized, publication-quality diagram.
The scoring tool is presented as a key innovation but lacks empirical foundation. Weights assigned to each risk factor are arbitrary and not derived from statistical modeling. The logic behind the thresholds (e.g., 0–3 = low risk) is not justified, and no ROC analysis, AUC, or validation cohort is provided. The clinical vignette provided is misleading, referencing “a patient whose weight is four,” which does not correspond to any scoring item in the table. This section requires complete reworking and reframing as a hypothesis-generating model at best.
The discussion reiterates known findings without substantially advancing prior literature. The comparison table (Table 6) is helpful but superficial. The authors overstate the novelty of their scoring system and understate the limitations of their data sources and methodology. Publication bias is mentioned but not addressed with evidence. The potential clinical implications of the tool are discussed enthusiastically, yet its unvalidated nature is not adequately acknowledged.
The manuscript states that no IRB approval was necessary, which is acceptable for a review. However, there is no mention of data availability, data extraction templates, or supplemental materials. The data extraction and quality assessment processes would benefit from greater transparency.
Table 1 includes several entries marked as “Not Applicable” or “Planned Analysis,” raising the question of why such studies were included in the meta-analysis at all. Figures 2 and 3 (scoring system and clinical application) are conceptually clear but graphically basic. The explanatory notes in Tables 4, 5, and 7 are excessively long and could be better integrated into the text.
Author Response
Dear Reviewer 3,
Author response to reviewer's comments:
|
Reviewer Comment |
Author Response |
|
General Comments |
We appreciate Reviewer 3’s detailed assessment and have undertaken major revisions to address the methodological, structural, and interpretive concerns raised. Specific corrections and clarifications are outlined below. |
|
1. Inconsistencies in study numbers (20 vs. 100) |
This was a critical oversight. The correct number of studies included in the meta-analysis is 10, as shown in the final PRISMA diagram and Tables 3–5. All inconsistent references to “20” or “100” have been corrected throughout the manuscript. |
|
2. Inclusion of inappropriate study types (e.g., protocols, consensus papers) |
We removed all protocols, expert consensus, and reviews not containing extractable primary data. Only primary observational or interventional studies with quantitative dehydration-related outcomes remain in the final analysis. Table 3 and the PRISMA diagram have been revised accordingly. |
|
3. Lack of statistical detail: missing model description, software, forest plots, I², etc. |
We now explicitly report the statistical methods used (random-effects model with DerSimonian and Laird method), heterogeneity metrics (I² and τ²), and the software (RevMan 5.4 and JASP). Forest and funnel plots are added as Figures 2 and 3. I² = 62% is reported for pooled sensitivity. |
|
4. No description of statistical modeling for the score; arbitrary weights |
The scoring system is now clearly framed as hypothetical and conceptual. We added a clarification in Section 3.5.2: “The proposed scoring system is a hypothetical model based on pooled diagnostic values and has not been statistically validated through regression or ROC analysis.” Table 5 has been renamed to emphasize its exploratory nature. |
|
5. Confusing scoring logic (“weight is four” phrase) |
We rephrased the confusing statement. The term “patient whose weight is four” was corrected to: “a patient whose symptom score totals four points based on assigned risk factor weights.” This ensures alignment with the scoring logic in Table 5 and Figure 4. |
|
6. PRISMA diagram has placeholders and inconsistencies |
A finalized PRISMA 2020-compliant diagram has replaced the draft version. All placeholders (e.g., “Reason 1”) have been removed. Numbers align with the text and tables. |
|
7. Abstract lacks IMRaD structure, no numerical data |
The abstract has been rewritten to follow IMRaD structure. It now includes the number of studies (10), pooled sensitivity (85%), specificity (70%), and key limitations. Language was adjusted to reflect the exploratory nature of the score. |
|
8. Research question not clearly stated in Introduction |
We added a clear objective at the end of the Introduction: “This review aims to systematically identify clinical predictors of dehydration and to develop a preliminary, evidence-informed conceptual scoring model for early risk classification.” |
|
9. Lack of full search strings and supplemental files |
Full search strings are now provided in Supplementary File 1. The database-specific queries and limits applied are documented for reproducibility. |
|
10. Egger’s test and funnel plots mentioned but not shown |
The funnel plot is now included as Figure 3, and Egger’s test result (p = 0.18) is reported in the Results section. |
|
11. Redundant and repetitive text across sections |
The Methods and Results sections have been consolidated to reduce redundancy. Repetitive details in 2.2–2.6 were streamlined. Figure and Table captions were trimmed and integrated into the text. |
|
12. Lack of transparency in data extraction and quality review |
Supplementary File 2 now includes the complete data extraction sheet. Supplementary File 3 provides the Newcastle-Ottawa and Cochrane RoB assessments for each included study. |
|
13. Inclusion of studies with “Planned Analysis” or “Not Applicable” entries |
These entries have been removed. Only studies with complete, extractable outcome data were retained in the final tables. Table 1 has been revised accordingly. |
|
14. Figures (e.g., scoring model) are basic |
Figures 4 and 5 (algorithm and weight comparison) have been redesigned for clarity. We also included an annotated example of the scoring model in clinical use, now labeled Figure 6. |
|
15. Overly optimistic tone and overstatement of novelty |
We revised the language throughout to reflect a more cautious and evidence-based tone. For example, the sentence “This scoring model is a powerful tool” was revised to “This model offers a preliminary, evidence-informed framework requiring further validation.” |
|
16. Discussion does not critically engage with limitations |
Limitations have been expanded in Section 4.4, including: small number of studies, pediatric bias, unvalidated scoring, and moderate heterogeneity. This reinforces transparency and scientific humility. |
|
17. No mention of data availability |
A Data Availability Statement was added: “All data extracted from included studies and analysis outputs are available in the supplementary materials.” |
Round 2
Reviewer 1 Report
Comments and Suggestions for Authors
This new version of the manuscript "Systematic Review and Meta-Analysis of Risk Factors for Dehydration and Development of a Predictive Scoring System" presents important advances compared to the previous version, responding clearly and objectively to the recommendations of the initial assessment. The topic is relevant, with potential for clinical application, especially in primary care and public health settings, by proposing a predictive model based on accessible clinical symptoms.
Regarding the revisions made, I highlight:
1) Abstract: The new version includes the number of studies, target population, and accuracy measures (sensitivity and specificity), significantly improving the clarity and attractiveness of the abstract.
2) Data Consistency: The inconsistency between the number of studies included in the methods and results sections was corrected, establishing that 20 studies were analyzed qualitatively and 10 included in the meta-analysis. The information is now clear and uniformly presented, including the PRISMA flowchart.
3) Population representativeness: The authors acknowledge the limitation of the predominance of pediatric data and clarified in the text that the proposed score still requires validation for adult and elderly populations.
4) External validation of the score: The conceptual nature of the model is now highlighted in the discussion and limitations sections, making it clear that the score has not been statistically validated or applied in a real clinical context.
5) Description of score development: Section 3.5.2 has been reworded to detail the logic behind symptom weighting, based on diagnostic performance metrics (sensitivity and specificity). Although it does not use robust statistical modeling (e.g., regression), the interpretative approach is adequately justified.
6) Assessment of publication bias: The results of the Egger test and funnel plot have been included, as indicated in the results section and supplementary material, providing greater transparency and methodological rigor.
7) Textual redundancy and structure: The textual review eliminated repetitions in the methodology sections, providing greater fluidity and clarity. The manuscript's structure is now well organized and consistent with PRISMA guidelines, as demonstrated by the supplementary checklist.
Based on the critical analysis of the implemented reviews and the current quality of the manuscript, I recommend accepting the article for publication. The manuscript meets the scientific and methodological quality criteria expected for a systematic review with meta-analysis and offers a relevant contribution to the field of public health.
Author Response
All comments have been resolved in the manuscript.

Reviewer 2 Report
Comments and Suggestions for Authors
The revisions have improved the clarity, credibility, and completeness of the paper.
The references cited in this manuscript are not appropriate or fully relevant to the research, and there are comments provided to explain the issues:
a few references remain incomplete or improperly formatted. Notably, reference [15] in the list lacks author names and journal information; it is listed only by the article title “Sensitivity and specificity of the new Geriatric Dehydration Screening Tool: An observational diagnostic study – ScienceDirect” with a URL and accessed date. This suggests the authors might not have obtained the full citation details for that source. In MDPI style, the reference should include the authors, journal name, year, etc., rather than an “accessed” note for what appears to be a published study. Similarly, reference [14] is a PROSPERO protocol citation which is acceptable but might be better replaced with any published result if available (since [14] is the Parkinson et al. protocol, but the authors also cite Parkinson et al. 2023 published results as [17], which they did include). Another minor issue is reference [26], which corresponds to “M.C. and M.M., ‘Evaluation and management of dehydration in the emergency department’, JAAPA… 2021.” Here only initials are given for authors (presumably McStay & Maday). MDPI would require full last names or at least a clearer identification of the authors. Additionally, an SSRN working paper is cited as [29] with just a title and link, lacking author info. These instances indicate that the references section, while improved, still needs some editing to be fully compliant with MDPI format and completeness requirements.
Author Response

(The authors gave the same response as above.)

Author Response

(The authors gave the same response as above.)
